# A Methodological Quality Evaluation of Meta-Analyses on Nursing Home Research: Overview and Suggestions for Future Directions

**DOI:** 10.3390/ijerph19010505

**Published:** 2022-01-03

**Authors:** In-Soo Shin, Juh-Hyun Shin, Dong-Eun Jang, Jiyeon Lee

**Affiliations:** 1Graduate School of Education, Dongkuk University, Seoul 04620, Korea; s9065031@dongguk.edu; 2College of Nursing, Ewha Womans University, Seoul 03760, Korea; 3School of Nursing, University of Texas at Austin, Austin, TX 78712, USA; dejang@utexas.edu; 4College of Nursing, Catholic University of Pusan, Seoul 43241, Korea; jylee@cup.ac.kr

**Keywords:** meta-analysis, nursing homes, evidence practiced nursing

## Abstract

(1) Background: The nursing home (NH) research field lacks quality reporting about meta-analyses (MAs), and most gradings of MA evidence are biased on analyzing the effectiveness of independent variables in randomized control trials. (2) Objectives: This study aimed to perform a critical methodological review of MAs in the NH research field. (3) Methods: We searched the articles from four databases (PubMed, MEDLINE, CINAHL, and PsycINFO) until 15th January 2021. We reviewed a total of 41 published review articles in the NH research field. (4) Results: The studies primarily fell into the following categories: medicine (17/41), nursing (7/41), and psychiatry or psychology (6/41); 36.6% of the reviewed studies did not use any validated MA guidelines. The lowest correctly reported PRISMA 2000 guideline item was protocol and registration (14.6%), and more than 50% of articles did not report risk of bias. Moreover, 78.0% of studies did not describe missing reports of effect size formula. (5) Discussion: NH researchers must follow appropriate and updated guidelines for their MAs in order to provide validated reviews, as well as consider statistical issues such as the complexity of interventions, proper grouping, and scientific effect-size calculations to improve the quality of their study. Future quality review studies should investigate more diverse studies.

## 1. Introduction

The fast-growing elderly population is creating serious problems globally, and providing appropriate care for nursing home (NH) residents is a priority issue [1]. More than 2200 studies were searched in the main search engines regarding oral and dementia care, falls, fractures, cognitive functioning, pain, restraints, infections, nutrition, mortality rates, hospital admissions, psychiatric problems, and some organizational factors in numerous health-related disciplines reporting different levels of evidence. NHs have some unique characteristics to consider, such as (a) resident- and organizational-level factors, (b) lengthy resident stays in specific NHs, and (c) randomization, which may not be possible, as most residents are vulnerable either physically or cognitively. Researchers in many disciplines have mainly studied NHs to improve resident care, including oral care, falls, fractures, cognitive functioning, pain, restraints, infections, dementia care, nutrition, mortality rates, hospital admissions, psychiatric problems, and some organizational factors in numerous health-related disciplines reporting different levels of evidence; however, it is very challenging to identify or assess NH care quality. NH researchers have been synthesizing results to translate their findings into practice for several decades. In the nursing discipline, evidence-based nursing practices integrate theory and employ clinical decision-making, judgement, and scholarly knowledge to derive the most effective and useful evidence related to specific elements of practice [1]. Transitioning nursing research to practice should be cautiously implemented with valid research conclusions derived from multiple sources.

The term meta-analysis (MA) was introduced in 1976 and is defined as organized and validated statistical processes used to compile research findings to answer specific research questions. MAs make it possible to achieve a more precise intervention effect considering the size, heterogeneity, and quality of studies with the highest levels of evidence [2]. The resulting evidence can be applied to nursing practices or to develop clinical guidelines [3]. Applying MAs in NH research offers many advantages without requiring extra data collection; for example, MAs may improve the researcher’s ability to conduct hypothesis testing, thus identifying patterns and outcomes, and can compile results with stronger estimates [4]. A traditional MA with a randomized control trial (RCT) is performed by synthesizing and combining relevant research articles to achieve the best and highest levels of answers for a specific research question [5]; however, the synthesis of observational studies has also been performed over the last 10 years in NH research. Researchers estimate the mean or effect size of intervention by statistically combining more than two research studies on the same topic [6]. It is important to use current scientific systematic reviews in order to find gaps in specific research areas, apply research-based evidence or guidelines into nursing practices, and implement policy [7]. Scholars have performed NH research using diverse research designs. However, most gradings of MA evidence are biased toward figuring out the effectiveness of independent variables of RCTs, and should now extend to an evaluation of its suitability to study populations, settings, and designs [4,8]. An RCT is considered the highest level of evidence, but RCTs may have less external validity despite their good internal validity [4]. For example, residents’ characteristics, government policies, regulations, and the financial operation of each NH are very different based on different geographical areas or countries. Additionally, it may be hard to secure a suitable sample size and study period due to the season, outbreak of infectious diseases, or residents’ deaths when applying RCTs in NH research [4]. The reporting tools of MAs with RCT designs include Assessing the Methodological Quality of Systematic Reviews 2 (AMSTAR 2) and Preferred Reporting Items for Systematic Reviews and Meta-Analyses 2020 (PRISMA 2020) [9], whereas MAs for observational studies use Meta-Analysis of Observational Studies in Epidemiology (MOOSE) [10]. No study has yet performed a methodological quality assessment of systematic reviews or MAs in NH research.

There are three types of MAs that can be used according to a study’s effect size and research design: (a) intervention MA, (b) measure-of-association MA, and (c) diagnostic-test-accuracy MA [8]. Intervention MA is the method used to achieve a conclusion about an intervention’s overall effects [11]. Measure-of-association MA refers to the method of synthesizing primary studies that scholars identified through correlation or structural-equation modeling between variables [12]. Diagnostic-test-accuracy MA refers to the method of synthesizing sensitivity and specificity from multiple studies to evaluate a diagnostic test’s performance [13]

Most review articles in the healthcare field have low [14] or moderate validity for use in evaluating methodological issues [15]. The quality of systematic reviews in 107 nursing journals was lower than that of orthodontic research or research about pain, possibly due to the scholars’ failures to adhere to the validated PRISMA instrument [16]. Concerns regarding systematic reviews’ scientific qualities have increased, especially in the nursing field [17]. Only 30 out of 107 Institute for Scientific Information (ISI) nursing journals required researchers to follow the PRISMA guideline for systematic reviews or MA articles [7]. Moreover, the registration of a systematic review is now required; however, there is no available information on how many journals require prior registration of their systematic review protocol on platforms such as the Prospective Register of Ongoing Systematic Review (PROSPERO). PROSPERO is a free, publicly accessible international registration database of prospectively registered systematic reviews in health and social care, operated through the University of York (Best Practice in Systematic Reviews: The Importance of Protocols and Registration The PLoS Medicine Editors).

The interpretation of MA results should be performed with close attention since the results may suffer from methodological study weakness [18]. The purpose of this study was to perform a critical methodological review of MAs in NH research.

## 2. Materials and Methods

This is a critical appraisal of MAs on NHs according to the PRISMA 2020 guideline’s 27 items.

### 2.1. Inclusion and Exclusion Criteria

Studies published from 1990 to 2020 that met the following inclusion criteria were selected. (1) We included all published articles in English about MAs in the NH research field. (2) We included any type of research design in any academic field beyond health services (nursing, medicine, dentistry, psychology, public health, rehabilitation, pharmacology, nutrition). (3) We also included studies with combined quantitative and qualitative designs. The exclusion criteria were as follows: (1) We excluded articles written in languages other than English. (2) We excluded articles based outside the NH setting (i.e., acute settings, long-term care hospitals, community-dwelling settings).

### 2.2. Information Sources and Search Strategy

We performed a systemic electronic search to evaluate published review articles in the NH research field, which we extracted from the four following scholarly health-related databases: PubMed, MEDLINE, CINAHL, and PsycINFO. The key words were “meta-analysis”, “systematic review”, “long term care”, and “nursing home”. We tailored the search strategy for each database. An example search for Pubmed is as follows: (“meta-analysis” [mh] OR “meta-analysis” [all] OR “meta-analyses” [all] OR “meta analysis” [all] OR “meta analyses” [all]) AND (“review” [pt] OR “systematic review” [all]) AND (“nursing home” [mh] OR “long term care” [all] OR “nursing home” [all]). We carried out the database search until 15 January 2021.

### 2.3. Study Selection and Data Extraction

We retrieved studies with the key words after the database search. Three reviewers independently screened the titles and abstracts of all articles from the initial search after excluding duplicity and non-NH settings. The reviewers obtained and examined the full texts of those articles passing the title and abstract screening. Three other reviewers resolved uncertainties or unclear methodologies through discussion. Two clinical experts in nursing and one expert in statistics coded and evaluated the clinical, methodological, and statistical parts together. Of the final included studies, we reviewed articles using a structured data-extraction strategy. Three team members independently extracted information from the included studies in a Word file, which included the title, first author, year of publication, study design, review protocol, synthesis method, and main findings. The kappa statistics value showed a high degree of agreement between the three reviewers in their decisions about the relevance of studies (kappa value = 0.89).

### 2.4. Planned Methods of Analysis

#### 2.4.1. Reporting of Epidemiological and Descriptive Characteristics

We presented each item from the PRISMA checklist by either the ratio or percentage of how many of the 41 PRISMA articles were properly followed [9] (see Table 1).

#### 2.4.2. Quality Assessment

We conducted the quality assessment (methodological systematic review) in compliance with the appropriate guideline. We evaluated a total of 41 articles on MAs in NHs by study design, the use of validated review tools, and kinds of tools. Secondly, we assessed whether the reviewed articles aligned with the 27-item PRISMA 2020 guideline (a validated checklist comprising of 27 items). The authors of this article checked each item using the Excel coding sheet and determined differences by agreement via regular e-mail and Zoom meetings. The PRISMA checklist did not fulfill some requirements. We also evaluated additional methodological and statistical issues based on the Meta-Analysis Reporting Standards (MARS) [6,58,59].

## 3. Results

Figure 1 presents the degree to which the reviewed articles adhered to the PRISMA checklist.

In this review, we screened a total of 1130 potentially relevant references identified through a search of bibliographic databases. After deleting duplicates and non-NH or non-long-term care facility settings (1057), the initial search yielded 73 publications. After screening abstracts, we excluded 32 references (30: systematic review only, 2: only qualitative studies). Finally, we reviewed and synthesized 41 publications.

Table 2 presents the summary of the reviewed articles. In total, 15 of 41 studies did not use any of the guidelines validated for MAs. The guidelines used in the remaining 26 studies were as follows: 17 studies used only the PRISMA guideline; 3 studies used only the MOOSE guideline; 3 studies used both the PRISMA guideline and Cochrane handbook simultaneously; 3 studies used both the PRISMA and MOOSE guidelines simultaneously.

Scholars mainly studied MAs in the NH research in the following areas: medicine (17/41 of reviewed studies) [20,25,26,28,36,38,39,40,47,48,49,50,51,52,54,57,60], nursing (7/41) [24,31,32,33,34,35,56], psychiatry or psychology (6/41) [22,30,44,45,55,61], public health (4/41) [21,27,37,46], and pharmacology (3/41) [23,42,53], followed by the areas of rehabilitation (1) [41], nutrition (1) [29], dentistry (1) [43], and nursing and dentistry (1) [19]. In the reviewed articles, some studies (12/41) examined the physical issues NH residents face, including falls, exercise, or rehabilitation issues. A few studies were conducted on NH placement, admission or mortality, death, nutrition and tube feeding, depression, oral health, cognitive function, pain management, infection, depression, or fragility. Only one study focused on the organizational characteristics of NH ownership. Most reviewed articles were classified as intervention MAs (21/41), measure-of-association MAs (21/41), and diagnostic-test-accuracy MAs (1/41). Measure-of-association represented various coefficients (including bivariate correlation coefficients and regression coefficients) that measured the strength and direction of relationships between variables [62]. These intensity or association measures could be explained in several ways, depending on the analysis. Measure-of-association in MAs assessed studies using methods such as correlation, regression, and structural-equation modeling between variables [12]. Future research should include more measure-of-association MAs in NH research. The Cochrane handbook emphasizes the intervention complexity or logic model, wherein various research types must be well classified and applied accordingly.

Among the PRISMA items, only six studies [19,21,22,23,24,25] correctly reported the item of “protocol and registration” (i.e., indicating whether a review protocol existed, if and where it can be accessed and if it was available, and providing registration information including the registration number). Only 21% of articles in the medical field reported a protocol [63]. Moreover, more than 50% of reviewed articles did not address the “risk of bias within studies” item, which was associated with former studies and was essential in calculating effect size [18,64].

Additionally, the majority of reviewed studies (32/41) did not offer an effect size variance formula [58], whereas high-impact factor journals effectively covered effect size calculations and statistical-analysis issues. The effect size variance formula is important as it tells us about normality and homogeneity assumptions [65]. Thus, concrete information on effect size computation should be reported in MAs [6,58]. Researchers used H (1/41), Q (18/41), and *I2* inconsistency (29/41) statistics for heterogeneity tests. Heterogeneity means the degree of difference in the results of each single research finding [66]. Through MAs, scholars calculate the heterogeneity index so they can understand the primary factors that impact individual studies’ effect sizes [67].

## 4. Discussion

This is the first study that scientifically assesses the methodological quality of systematic reviews used for MA studies in NH research. The scientific quality evaluation of MA studies on NHs provides reliable guidance on nursing policies and practices that improve outcomes for residents based on assessments of the quality of evidence from prior MA studies. The major benefits of MA are that it has strong statistical power with effect size and offers conclusions that individual studies often cannot reach [26]. The effect size variance formula is important as it tells us about normality and homogeneity assumptions [65]. Thus, concrete information on effect size computation should be reported in MAs [6,58]. Scholars should investigate assumptions and report scientific research findings in a scientific way to ground more valid, superior, evidence-based nursing practices for use in NH research. Scholars have studied NH research for several decades, and scientific synthesis is required to address current nursing issues, usually focusing on quality-of-care issues including pressure ulcers, falls, activities of daily living, and others.

The studies on NH that have been synthesized include various study designs, such as RCTs and observational, prevalence, diagnostic test accuracy, and association studies. Scholars must consider the most appropriate method for exploring validated reviews in NH research. Usually, NH research is performed using an observational design since researchers can investigate the process of care and reasons for missed care [19]. The JBI tool includes experimental and non-experimental designs of methodological reviews, but other critical appraisal tools are not often used due to the permission requirements [68]. Scholars mainly use PRISMA for reviews that assess the effects of interventions, prognoses, diagnoses, and prevalence [7]. However, some of the reviewed studies used PRISMA (which should not be used for observational studies) despite not having RCTs [20,27,28,29]. Unintentional inadvertence may occur when reporting reviews without adhering to valid review guidelines. For example, the APA MARS is a good guideline for association and relational MA in addition to non-RCT systematic reviews and MAs in NH fields. The APA Working Group on Journal Article Reporting Standards developed the APA MARS in 2008 after the group synthesized the Quality of Reporting of MA statement, PRISMA, MOOSE, and Potsdam Consultation on MA [58]. The Newcastle–Ottawa scale has also been used when reviewing observational studies, including case–control, prospective, and cohort studies [69]. This scale has three areas of focus (participants of study, comparison between case/controls, and evaluation of outcomes) linked with the quality concept, and it is recommended to have at least two reviewers, as the review involves quite subjective characteristics [69]. The MOOSE is a reporting guide for MAs on observational studies of epidemiology, which improves the reporting on MAs of observational studies [10]. The Strengthening the Reporting of Observational Studies in Epidemiology (STROBE) initiative developed recommendations on what should be included in an accurate and complete report of an observational study. STROBE was developed to improve the reporting quality of observational studies [70]. Regarding quality assessment tools, one of the examined studies used Consolidated Standards of Reporting Trials (CONSORT) [4] to measure the methodological quality of individual studies included in the MA research. However, CONSORT is a set of recommendations related to how to report on individual studies in randomized trials, and is not actually a quality assessment tool to establish internal validity in MAs. Even though meta research is considered the highest scientific level of evidence, many potential sources of bias exist [71] including selection, performance, attrition, and detection bias, which affect internal validity. Therefore, scholars must use valid and standardized methodological quality assessment tools for the individual studies included in MA research to ensure the quality of synthesized results.

The ROB 2 is the gold standard used to evaluate the quality of RCT biases [72]. Most of the reviewed articles (11/41) [21,22,23,24,26,28,32,35,53,56,61] used the ROB, but not the ROB 2. The ROB 2 is a reorganized form of the Cochrane ROB tool, developed by the same team [73]. The tool allows researchers to simply decide if a bias exists in the reviewed research, and can be used to evaluate bias in particular result findings beyond the RCT design [73]. More unreliably, one paper used the ROB but did not use an RCT [28]. Effect size refers to the numerical index of findings and is regarded as an outcome variable in MA research [8]. Effect size usually consists of d (odds ratio) for experimental designs and R for correlational studies [8]. A specific process with calculations should be provided when reporting on effect size. Although most reviewed articles are intervention MA, the use of diagnostic MA with test accuracy is increasing in healthcare research [8]. One of the reviewed articles [30] reported on the proportion of residents with depression, which is categorized as diagnostic MA. We suggest performing diagnostic MA using the appropriate standard in order to have accurate data on the prevalence or incidence of specific disease, infectious disease, or the progress of dementia. Based on this, MA results can be applied for developing sensitive instruments for use in NH populations’ [31].

The intervention complexity of studies (intervention number and the interaction among interventions in a system’s structure) should be considered in the future, as a mixture of interventions should be applied to NH residents with a more accurate intervention estimation [74]. The interventions that were conducted to improve residents’ physical, psychological, and cognitive functions included fall prevention, nutrition, oral care, ambulant, feeding, exercise, and so on. Scholars should clearly define the research objectives, population to be studied, and scope of treatments when performing MAs to answer the research questions [74]. NHs have some unique characteristics to consider, such as (a) resident- and organizational-level factors, (b) lengthy resident stays in specific NHs, and (c) randomization. NH researchers have been synthesizing results to translate their findings into practice for several decades. In the nursing discipline, evidence-based nursing practices integrate theory and include clinical decision-making, judgement, and scholarly knowledge to achieve the most effective and useful evidence for specific elements of practice [1]. Researchers conduct and share research results on nursing practices, education, and administration. Transitioning nursing research to practice should be cautiously implemented with valid research conclusions derived from multiple sources.

Lastly, the appropriate registration of reporting guidelines before publication should be performed to improve the methodological quality using PROSPERO. Protocol registration is not a mandatory requirement in MAs, but it should be emphasized to decrease duplication among studies, and it is also supported by the National Institute for Health Research (NIHR) and PROSPERO [63,75,76,77].

### Limitations

This study has some limitations, in that it did not have a PROSPERO registration number and only included articles published in English. Future studies should include more diverse research around the world to represent global citizens.

## 5. Conclusions and Implication

MA is one of the best scientific methods for dealing with big data. MA can be used to integrate large data into NH research, contributing to the open-science movement in research [78]. In the era of evidence-based nursing practices, NHs should prioritize deriving accurate and valid judgements for nursing practices. Thus, choosing and applying the appropriate and most conforming review tool is an important issue when reviewing articles. Researchers should use appropriate, updated quality assessment tools reflecting the study’s research design and objectives, and must consider the complexity of interventions, proper grouping, and scientific effect size calculations.

## Figures and Tables

**Figure 1 ijerph-19-00505-f001:**
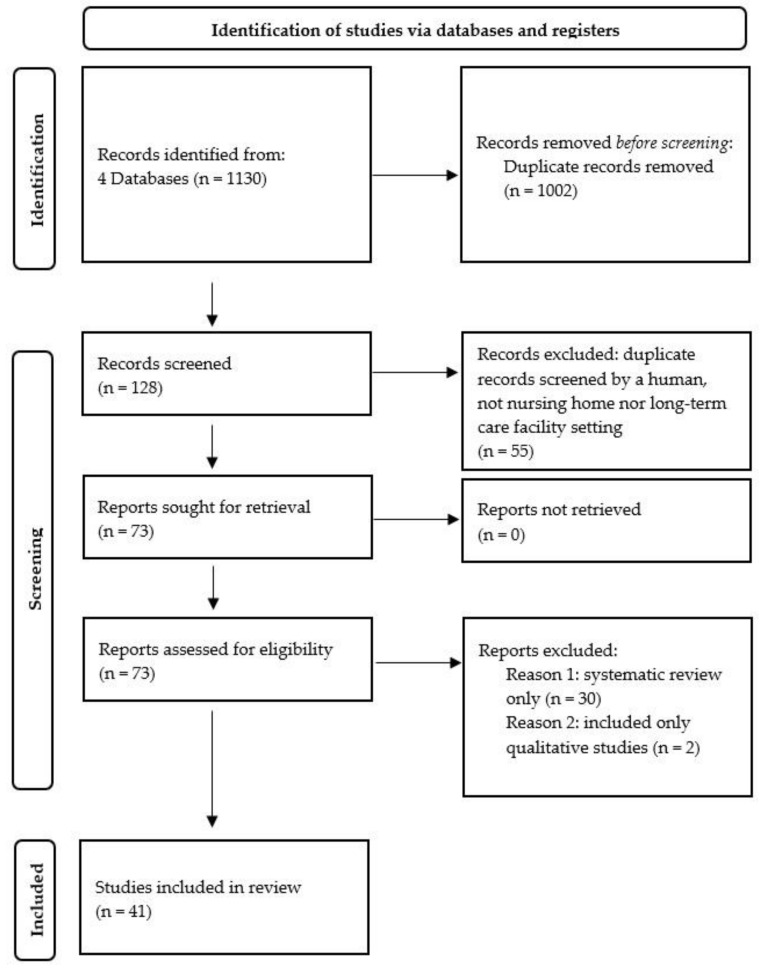
Flow diagram of the review selection on nursing home research.

**Table 1 ijerph-19-00505-t001:** PRISMA Checklist.

	PRISMA Item Number																										
1st Author, Year	Title 1	Structured Summary	Rationale	Objectives	Protocol and Registration	Eligibility Criteria	Information Sources	Search	Study Selection	Data Collection Process	Data Items	Risk of Bias in Individual Studies	Summary Measures	Synthesis of Results	Risk of Bias Across Studies	Additional Analyses	Study Selection	Study Characteristics	Results of Individual Studies	Results of Individual Studies	Synthesis of Results	Risk of Bias Across Studies	Additional Analysis	Summary of Evidence	Limitations	Conclusions	Funding
Hoben et al., 2017 [19]	O	O	O	O	O	O	O	O	O	O	O	O	O	O	O	O	O	O	O	O	O	O	O	O	O	O	O
Veronese et al., 2015 [20]	O	O	O	O	▲	O	O	O	X	O	O	X	X	O	O	O	O	O	O	O	O	X	O	O	O	O	O
Brugnolli et al., 2020 [21]	O	O	O	O	O	O	O	O	O	O	O	O	X	O	O	X	X	O	O	O	O	O	O	O	O	O	O
Fornaro et al., 2020 [22]	O	O	O	O	O	O	O	O	O	O	O	O	O	O	O	O	O	O	O	X	O	O	O	O	O	O	O
Kua et al., 2019 [23]	O	O	O	O	O	O	▲	O	O	O	O	X	O	O	O	O	O	O	O	X	O	X	O	O	O	O	X
Vlaeyen et al., 2015 [24]	O	O	O	O	O	O	▲	O	O	X	O	O	X	O	O	O	O	O	O	X	O	O	O	O	O	O	O
Zhang et al., 2018 [25]	O	O	O	O	O	O	O	O	O	O	O	O	O	O	O	O	O	O	O	O	O	O	O	O	O	O	O
Silva at el., 2013 [26]	O	O	O	O	X	O	▲	O	O	O	O	X	X	O	X	X	O	O	O	X	O	X	X	O	X	O	X
Costa et al., 2016 [27]	O	O	O	O	X	O	O	O	O	O	O	O	X	O	O	X	O	O	O	X	O	O	X	O	O	O	O
Lan et al., 2017 [28]	O	O	O	▲	X	O	▲	O	X	O	O	O	X	O	O	O	O	O	O	X	O	O	O	O	O	O	X
Cereda et al., 2016 [29]	O	O	O	O	X	O	▲	O	O	O	O	X	X	O	X	O	O	O	O	O	O	X	O	O	O	O	O
Mitchell et al., 2011 [30]	O	O	O	O	X	X	O	O	O	O	X	X	X	O	X	O	X	▲	O	O	O	X	O	O	O	O	O
Gaugler et al., 2007 [31]	O	O	O	O	X	O	▲	O	O	O	O	X	X	O	O	O	O	O	O	O	O	X	O	O	O	O	O
Lee et al., 2017 [32]	O	O	O	O	X	O	▲	O	O	O	O	O	O	O	X	X	O	O	O	X	O	X	X	O	X	O	O
Li et al., 2015 [33]	O	O	O	O	X	O	▲	O	X	X	X	X	X	O	X	X	O	O	O	O	O	X	X	O	O	O	X
Knopp-Sihota et al., 2016 [34]	O	O	O	O	X	O	O	O	O	O	O	O	X	O	X	X	O	O	O	X	O	X	O	O	X	O	O
Aliyu et al., 2017 [35]	O	O	O	O	X	O	▲	O	O	O	O	O	X	O	O	O	O	O	O	X	O	O	O	O	O	O	X
Deandrea et al., 2013 [36]	O	O	O	O	X	O	▲	O	O	O	O	X	X	O	X	X	O	O	O	X	O	X	X	O	X	O	O
Petrignani et al., 2015 [37]	O	O	O	O	X	O	▲	X	O	O	O	X	X	X	X	X	O	O	O	O	O	X	X	O	O	O	O
Comondore et al., 2009 [38]	O	O	O	O	X	O	O	O	O	O	O	X	O	O	O	O	O	O	O	O	O	O	O	O	O	O	O
Kojima, 2015 [39]	O	O	O	O	X	O	▲	O	O	O	O	X	X	O	X	X	O	O	O	O	O	X	X	O	O	O	X
Kojima, 2018 [40]	O	O	O	O	O	O	▲	O	O	X	O	X	X	O	O	X	O	O	O	O	O	O	O	O	O	O	X
Crocker et al., 2013 [41]	O	O	O	O	X	X	X	X	X	X	X	X	X	X	X	X	O	O	O	O	O	O	O	O	O	O	O
Wallerstedt et al., 2014 [42]	O	O	O	O	X	O	O	O	O	O	O	O	X	O	X	O	O	O	O	O	O	X	O	O	O	O	O
Sjögren et al., 2016 [43]	O	O	O	O	X	O	O	O	O	O	X	X	X	O	X	O	O	X	O	O	O	X	X	O	O	O	O
Robertson et al., 2017 [44]	▲	O	O	O	▲	O	▲	O	X	O	X	X	X	X	X	X	O	O	O	O	O	X	X	O	O	O	O
Toot et al., 2017 [45]	O	O	O	O	X	O	▲	O	O	X	X	X	X	X	X	X	O	O	O	O	O	X	X	O	O	O	X
Jutkowitz et al., 2016 [46]	O	O	O	O	▲	O	O	O	O	O	O	O	O	O	O	O	O	O	O	O	O	X	O	O	O	O	O
Smith et al., 2016 [47]	O	O	O	O	X	O	O	O	O	O	O	X	O	X	X	O	O	O	O	X	O	X	X	O	O	O	O
Kuys et al., 2014 [48]	O	O	O	O	X	O	O	O	X	O	O	X	O	O	O	O	O	O	O	O	O	O	X	O	O	O	X
Hedrick et al., 1989 [49]	O	O	O	O	X	X	▲	O	X	X	X	X	X	O	X	X	O	O	O	O	O	X	X	O	O	O	X
Silvia et al., 2019 [50]	O	O	O	O	O	O	O	O	O	O	O	O	O	O	X	O	O	X	O	O	O	X	X	O	O	O	X
Gulka et al., 2020 [51]	O	O	O	O	X	X	▲	O	X	X	X	X	X	O	X	X	O	O	O	O	O	X	X	O	O	O	X
Zhang et al., 2019 [52]	O	O	O	O	O	O	O	O	O	O	O	O	O	O	X	O	O	X	O	O	O	X	X	O	O	O	X
Shaun et al., 2019 [53]	O	O	O	O	O	O	▲	O	O	O	O	X	X	O	X	X	O	O	O	X	O	X	X	O	O	O	X
Shen et al., 2019 [54]	O	O	O	O	O	O	▲	O	O	O	O	X	X	O	X	X	O	O	O	X	O	X	X	O	O	O	X
Prins et al., 2020 [55]	O	O	O	O	▲	O	O	O	O	X	O	O	O	O	O	O	O	O	O	X	O	O	O	O	O	O	O
Cao et al., 2017 [56]	O	O	O	O	▲	O	O	O	X	O	O	O	O	O	O	X	X	O	O	X	O	O	O	O	O	O	O
Lan et al., 2017 [57]	O	O	O	O	X	O	▲	O	O	O	O	O	X	O	X	X	O	O	O	O	O	X	X	O	O	O	X

*Note*. O: well-followed; ▲: followed but not completed well; X: did not follow.

**Table 2 ijerph-19-00505-t002:** Summary of Included Studies.

1st Author, Year Field	Included Study Design (Numbers)	Review Design	Synthesis Method	Findings
Hoben et al., 2017 [19] Nursing, dentistry	Qualitative and Quantitative (45)	PRISMA and MOOSE guideline	- A thematic analysis - Random-effects models with STATA 13.1 METAPROP module - Freeman–Tukey double arcsine transformation - I2	Barriers: residents resisting care, care providers’ lack of knowledge/education/training in providing oral care, general difficulties in providing oral care, lack of time, general dislike of oral care, Fall prevention interventions decreased the lack of staff.
Veronese et al., 2015 [20] Medicine	Observational cohort studies (36)	PRISMA and MOOSE guideline	- DerSimonian–Laird random-effects model - Cochrane I2 statistics - Stratified analysis - Meta-regression analysis - Publication bias assessed	Being underweight had a significant effect on mortality caused by infections.
Brugnolli et al., 2020 [21] Public Health	RCT, quasi-experimental design study (12)	Cochrane handbook, PRISMA	- I2 statistics - Random or fixed effect model	Educational programs and other supplementary interventions should be effective, but the heterogeneous operative definition of physical restraints can make data generalization difficult.
Fornaro et al., 2020 [22] Psychiatry	Either naturalistic studies or interventional studies (36)	PRISMA and MOOSE	- I2 statistics - Cochran’s Q test	Pooled prevalence rate of major depressive disorder was 18.9%.
Kua et al., 2019 [23] Pharmacology	RCT (41)	PRISMA	- Cochran’s Q test - Random effects model - I2 statistics - Subgroup analysis	Deprescribing interventions significantly reduced the number of residents with potentially inappropriate medications by 59%.
Vlaeyen et al., 2015 [24] Nursing	An original or a priori secondary analysis of individual-level or cluster RCT (13)	PRISMA	- Random effects approach - I2 - Subgroup analysis - Sensitivity analysis	There was no significant effect of the intervention on fallers or falls.
Zhang et al., 2018 [25] Medicine	Prospective cohort studies (6)	MOOSE	- I2 statistics - Cochran’s Q test	Sarcopenia showed positive association with a risk for all-cause mortality.
Silva at el., 2013 [26] Medicine	RCT (12)	PRISMA	- Random effects model - I2	Exercise has a preventive effect on falls. This effect of mixing several types of exercises was stronger for 1–3 months or for more than 6 months, with a frequency of at least 2–3 times per week.
Costa et al., 2016 [27] Public Health	Retrospective cohort studies (26)	PRISMA	- Random effects model - I2 - Stratified analysis - RevMan version 5.2	Determinants of nursing home versus hospital death: multidisciplinary in-home palliative care, preference for home death, cancer diagnoses, early referral to palliative care, not living alone, presence of a caregiver, and the caregiver’s coping skills.
Lan et al., 2017 [28] Medicine	Comparative analysis, cross-section, case–control, prospective, retrospective design (13)	PRISMA	- RevMan version 5.3 - Q and I2 test - Random-effect model (DerSimonian and Laird method) - Fixed-effects model (Mantel–Haenszel method) - Z-test	Lower levels of hemoglobin and creatinine were shown in tube-fed patients.
Cereda et al., 2016 [29] Nutrition	Any type of non-interventional trial (240)	PRISMA	- DerSimonian–Laird random-effect model - Cochran’s Q and I2 statistics - Meta-regression - Stratified analysis - STATA 13.1	Prevalence of malnutrition: community 3.1%, outpatients 6.0%, home-care services 8.7%, hospital 22.0%, nursing homes 17.5%, long-term care 28.7%, rehabilitation/sub-acute care 29.4%.
Mitchell et al., 2011 [30] Psychology	Any Studies Design (22)	N/A	- Random effects model - Sensitivity analysis - Statsdirect, Stata10 - Bayesian curve analysis	Practice/community nurses and hospital nurses correctly identified 26.3% and 43.1% of depressed and 94.8% and 79.6% of the non-depressed people, respectively. Nursing home nurses correctly identified 45.8% of people with depression and 80.0% of the non-depressed.
Gaugler et al., 2007 [31] Nursing	Observational Studies (12)	N/A	- Random effect model - Cochran’s Q-statistic - Subgroup analysis - Sensitivity analysis - Meta-regression	Significant predictors of nursing home admission were three or more activities of daily living dependencies, cognitive impairment, and prior nursing home use.
Lee et al., 2017 [32] Nursing	Randomized Controlled Trials (RCT) (21)	N/A	- Review Manager, v.5.3 - Random effects model - Cochrane Q and I2 statistics - Mantel–Haenszel, inverse variance methods - Subgroup analysis - Sensitivity analysis	Exercise had a preventive effect on the fall rate. Combined exercise effect with other fall interventions was stronger on the rate of falls and number of fallers. Exercise interventions resulted in reduced rate of falls.
Li et al., 2015 [33] Nursing	RCT or non-RCT (16)	PRISMA	- MetaView Review Manager Version 5.2 - Random effect model - I2	No significant improvement in the short-term effects of music therapy.
Knopp-Sihota et al., 2016 [34] Nursing	Controlled Trials (14)	The Cochrane Collaboration, PRISMA	- The Cochrane Collaboration software program - Random effect model - Fixed-effect model - I2 - Subgroup analysis	Non-analgesic treatment and control groups showed no statistical differences.
Aliyu et al., 2017 [35] Nursing	Observational studies (12)	MOOSE	- Random effects model - I2 - Subgroup analysis - Sensitivity analysis	Prevalence for MDR-GNB colonization: 27%.
Deandrea et al., 2013 [36] Medicine	Prospective study design (24)	The Cochrane Collaboration	- RevMan version 4.3.2 - Random effect models - Sensitivity analysis - x2 test	Nursing home residents: history of falls, walking aid use and moderate disability. Hospital inpatients: history of falls.
Petrignani et al., 2015 [37] Public Health	Observational studies (58)	N/A	- Random effects model - SAS 9.3 and CMA	Transmission was associated with bedside care and exposure to vomit.
Comondore et al., 2009 [38] Medicine	Observational studies and RCT (82)	N/A	- Random effects models - x2 test and I2 - Meta-regression random effects model - Sensitivity analysis - Subgroup analysis	Not-for-profit facilities delivered higher-quality care than did for-profit facilities: more or higher staffing quality, lower pressure ulcer prevalence.
Kojima, 2015 [39] Medicine	Cross-sectional or observational studies (9)	N/A	- Cochran’s Q statistic - Random effects model - I2 - Fixed-effects model - Publication bias (Egger and Begg tests) - Stata IC 13, StatsDirect	Prevalence of frailty and prefrailty was 52.3% and 40.2%, respectively.
Kojima, 2018 [40] Medicine	Original longitudinal studies (5)	MOOSE guideline	- Mantel–Haenszel method - Cochran’s Q test - I2 - Fixed-effects model - Random-effects model - Publication bias (Egger and Begg tests) - Review Manager 5.2, StatsDirects version 2.8 - Subgroup analysis	Both frailty and prefrailty predicted nursing home placement significantly.
Crocker et al., 2013 [41] Rehabilitation	RCT (13)	N/A	- Random-effects models	ADL independence improved by 0.24 in standard units.
Wallerstedt et al., 2014 [42] Pharmacology	RCT and non-RCT (12)	The Regional Health Technology Assessment Centre in the Region Västra Götaland, Sweden	- RevMan version 5.1 - Mentel–Haenszel method in a random-effects model - I2 - Sensitivity meta-analysis	Medication review does not reduce mortality or hospitalization.
Sjögren et al., 2016 [43] Dentistry	RCT (5)	N/A	- Mantel–Haenszel method random-effects model - Fisher exact test	Dental personnel given oral care interventions decreased mortality from healthcare-associated pneumonia. Nurse-given oral care interventions showed no statistically significant difference.
Robertson et al., 2017 [44] Psychiatry	Quantitative studies (17)	PRISMA	- Stats direct version 3 - DerSimonian Laird method based on a random effects model	No difference between global proxy rated quality of life.
Toot et al., 2017 [45] Psychology	RCT, cohort studies, epidemiological studies, case–control studies, systematic reviews, descriptive studies (26)	PRISMA	N/A	Risk factor of nursing home placement: impairments in activities of daily living.
Jutkowitz et al., 2016 [46] Public health	RCT (19)	N/A	- Knapp–Hartung random effects model - I2 statistics - Sensitivity analysis	No strong evidence to support effectiveness of care-delivery interventions on managing agitation and aggression.
Smith et al., 2016 [47] Medicine	Quantitative and qualitative studies (24)	PRISMA	- I-consistency - Subgroup analysis	Prevalence of musculoskeletal pain was 30.2%.
Kuys et al., 2014 [48] Medicine	Any quantitative design studies (34)	N/A	- Fixed and random effects model - FMetafor package version 1.7-0 - Cochran’s Q test - Meta-regression	No association between gait speed and covariates. General-pace gait speed was 0.475 m/s and maximal pace speed was 0.672 m/s.
Hedrick et al., 1989 [49] Medicine	RCT and quasi experimental studies (13)	N/A	- Woolf’s method - Loglinear modeling - Logistic regression - Odd man out method	Small beneficial effect of home care.
Silvia et al., 2019 [50] Medicine	Qualitative and Quantitative (16)	PRISMA	- Random effects model - Meta-regression	EOL conversations between health care professionals and family had a positive effect on family’s decision to limit or withdraw life-sustaining treatments.
Gulka et al., 2020 [51] Medicine	RCT (36)	N/A	- Random effects model - I2 statistics - Subgroup analysis	Fall prevention interventions decreased the number of falls, fallers, and recurrent fallers.
Zhang et al., 2019 Medicine [52]	Any quantitative design studies (14)	N/A	- Cochran’s Q test - Random effects model - I2 statistics - Subgroup analysis - Sensitivity analysis	Residents with frailty were at an increased risk of mortality compared to those without frailty.
Shaun et al., 2019 [53] Pharmacology	RCT and Cross-sectional studies and retrospective studies (42)	N/A	- Random effects model - I2 statistics	Pharmacist-led services reduced the mean number of falls among residents.
Shen et al., 2019 [54] Medicine	All kinds of studies (16)	PRISMA	- Random effects model - I2 statistics - Cochran’s Q test	Prevalence of EWGSOP in women and men was 46% and 43%, respectively. Malnutrition was the independent factor of EWGSOP.
Prins et al., 2020 [55] Psychology	Experimental research design study (27)	N/A	- I2 statistics - Cochran’s Q test - Random effects model	Sensory stimulation improved nocturnal behavioral restlessness in terms of both sleep duration and continuation. The effect on the number of awakenings, RAR, and daytime sleep was not significant.
Cao et al., 2017 [56] Nursing	RCT (9)	Cochrane handbook, PRISMA	- I2 statistics - Cochran’s Q test	Exercise had no effect on fall prevention in nursing home residents.
Lan et al., 2017 [57] Medicine	RCT (10)	PRISMA	- Fixed-effects (Mantel–Haenszel) - Random effects (DerSimonian and Laird) models - Subgroup analysis - Meta-regression - Cochran’s Q-statistic - I2	Physical restraint use was significantly lower in the experimental (education) group.

*Note*. MOOSE: Meta-analysis of Observational Studies in Epidemiology; PRISMA: Preferred Reporting Items for Systematic Reviews and Meta-Analysis; N/A: not available.

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
