# Peer review of "A Methodological Quality Evaluation of Meta-Analyses on Nursing Home Research: Overview and Suggestions for Future Directions"

_ijerph, 2022, doi:10.3390/ijerph19010505_

Round 1

Reviewer 1 Report

The study gives greater visibility to the existing problem of the low methodological quality of research conducted in nursing homes. It is an issue that needs to be addressed. Nevertheless, I would like to make some recommendations about the manuscript:

  • Line 28. Where does this data come from? Please indicate the reference, as I doubt that it corresponds to number 1.
  • Line 33. Please explain in a few words why randomisation is sometimes difficult in NHs.
  • Line 52. Consider replacing "effect of" with "effect size of".
  • Lines 66-67. It is necessary to include the PRISMA 2020 statement reference.
  • Lines 69-70. Please add some current examples of studies.
  • Lines 84-85. It would be interesting to add how many journals require prior registration of the systematic review protocol on platforms such as PROSPERO.
  • Line 91. No mention is made of the design of this study, so this should be added. Although it is not exactly a systematic review, given the characteristics of the study, it should be noted that many aspects of its content meet or can meet most of the items of the PRISMA 2020 checklist. It would be desirable for it to meet all possible quality criteria that could be applied to such a study. For example, information could be added to meet items 2, 12, 16b, 23b and 23c.
  • Line 92. Please consider incorporating study publication date limits, where the studies were applied (NHs) and patients (residents with long term care) as criteria.
  • Line 93. When "various" academic fields are mentioned, are they only from the health field or are other external fields included? It would be interesting to specify this.
  • Line 95. Please consider replacing "studies" with "designs".
  • Line 122 (Table 1). A legend is needed in the table to clarify the circles, crosses and triangles. This version of the manuscript does not contain such data.
  • Lines 100-101. Please include the equations used for each database search as supplementary material to allow replication of the study.
  • Lines 103-104. Although it does not alter the final result of the selected articles, in the example equation, specifically in "("nursing home" [mh] OR "long term care" [all] OR "nursing home" [all])", may have led to articles unrelated to NHs (due to the inclusion of the term "long term care" together with the boolean operator OR).
  • Lines 124-125 (Figure 1). Please use PRISMA 2020 flow diagram for new systematic reviews which included searches of databases and registers only.
  • Line 130. Replace "PRISMA guideline" with "PRISMA 2020 guideline".
  • Line 137. It must be a typo, but both Figure 1 and Table 2 list 41 studies, while the text lists 39. Moreover, there is no mention of Figure 1 in the text.
  • Lines 138-143. This part of the text should be restructured and amended. (1) Indicate the number of studies in number format or text, but do not mix both nomenclatures. (2) Line 140 and then lines 142-143 refer to studies using both PRISMA and MOOSE, so it seems to be repeated information (although, strangely, the number of studies differs in each sentence). (3) Similarly, the same information seems to be repeated for PRISMA in lines 139 and 142. (4) In lines 141-142, when RCTs are mentioned, it would be desirable to move this information to the end of the paragraph, as it is a different type of data from what is observed in the paragraph.
  • Line 145 (Table 2). For each study listed in the table, please insert the corresponding number in the reference section.
  • Line 158. Please provide examples of measure-of-association in MAs.
  • Lines 165-167 and 172-174. Consider moving this information to Discussion.
  • Lines 195-196. This information is either confusing or I have misunderstood it. The PRISMA 2020 statement explains that it can be applied to different types of systematic reviews (including those that only review other systematic reviews).
  • Line 199. Some studies have assessed quality using ROB instead of ROB 2 because were published before ROB 2 was launched. Please specify which studies were evaluated using ROB when ROB 2 was already available.
  • Line 204. "d" for Odds Ratio or Cohen's d?
  • Lines 213-234. Many of the future recommendations focus on researchers. However, scientific journals can also contribute to the publication of research with higher methodological quality (requirement to use reporting guidelines from EQUATOR Network, prior publication of the protocol in prospective registers such as PROSPERO...). Please add these contributions.

I hope you find these comments useful in improving this article.
Best regards.

Author Response

December, 15, 2021

Executive Editor, International Journal of Environmental Research and Public Health

RE: Revision of Manuscript ID: ijerph-1492274 entitled "A Methodological Quality Evaluation of Meta-Analyses on Nursing Home”

Research: Overview and Suggestions for Future Directions"

To whom it may concern:

Attached please find an electronic copy of a first revision of Manuscript ID: ijerph-1492274 entitled "A Methodological Quality Evaluation of Meta-Analyses on Nursing Home,” reviewed for publication in the International Journal of Environmental Research and Public Health. In your editorial decision letter, you observed that the reviewers found considerable merit in the manuscript, but a number of issues required attention. You invited a revision and resubmission of the manuscript. We found your comments and those provided by the reviewers to be extremely helpful, and we hope you will find the revised manuscript to be substantially improved. Below, we provide a detailed point-by-point description of our response to each issue the reviewers raised. 

We believe we have addressed each issue and hope you find the revisions to be satisfactory. We look forward to hearing from you. Thank you for your time and consideration.

Reviewers' comments (Reviewer 1)

Response to the comments

Page and Lines

ï¾·  Where does this data come from? Please indicate the reference, as I doubt that it corresponds to number 1.

Thank you for your comment. We explained how we got this data from the search engine. More than 2,200 studies had been searched in the main search engines regarding oral and dementia care, falls, fractures, cognitive functioning, pain, restraints , infections, nutrition,

p.1 

Line 32-34

ï¾·  Please explain in a few words why randomisation is sometimes difficult in NHs.

Thank you for your comment. We explained why randomization is difficult in NHs. 

=>and (c) randomization, which may not be possible as most of residents are vulnerable either physically or cognitively with ethical issue. Researchers in many disciplines have mainly studied NHs to improve resident care including oral care, falls, fractures, cognitive functioning, pain, restraints, infections, dementia care, nutrition, mortality rates, hospital admissions, psychiatric problems, and some organizational factors in numerous health-related disciplines reporting different levels of evidence; although, it is very challenging to identify or assess NH care quality.   

p.1 

Line 38-42

ï¾·  Consider replacing "effect of" with "effect size of".

Thank you for your comment. We replaced the term.

=>effect size of  

p.2 

Line 63

ï¾·  It is necessary to include the PRISMA 2020 statement reference.

Thank you for your comment. We included the reference. =>Preferred Reporting Items for Systematic Reviews and Meta-Analyses 2020 (PRISMA 2020) [9],

p.2 

Line 78

ï¾·  Please add some current examples of studies.

Thank you for your comment. There is no study have examined the methodological quality assessment of systematic reviews or MAs in NH research. Therefore, we mentioned about that.

p.2 

Line 80

ï¾·  It would be interesting to add how many journals require prior registration of the systematic review protocol on platforms such as PROSPERO.

Thank you for your comment. We described this point. 

=>Moreover, the registration of systematic review is required recently, however, there is no reported information how many journals require prior registration of the systematic review protocol on platforms like PROSPERO. Prospective Register of Ongoing Systematic Review (PROSPERO) is a publicly free accessible international registration database of prospectively registered systematic reviews in health and social care through the University of York (Best Practice in Systematic Reviews: The Importance of Protocols and Registration The PLoS Medicine Editors). 

p.3 

Line 96-103

ï¾·  No mention is made of the design of this study, so this should be added. Although it is not exactly a systematic review, given the characteristics of the study, it should be noted that many aspects of its content meet or can meet most of the items of the PRISMA 2020 checklist. It would be desirable for it to meet all possible quality criteria that could be applied to such a study. For example, information could be added to meet items 2, 12, 16b, 23b and 23c.

Thank you for your comment. We stated this point. 

=>This is the critical appraisal of Meta-Analyses on nursing homes according to the PRISMA 2020 guideline’s 27 items.

p.3 

Line 108

ï¾·  Please consider incorporating study publication date limits, where the studies were applied (NHs) and patients (residents with long term care) as criteria.

Thank you for your comment. We stated inclusion and exclusion criteria in more detail.  

=> Studies published from 1990 to 2020 that met the following inclusion criteria were selected. 1) We included all published articles in English about MAs in NH research. 2) We included any type of research in designs in any academic field beyond health services (nursing, medicine, dentistry, psychology, public health, rehabilitation, pharmacology, nutrition). 3) We also included studies with combined quantitative and qualitative de-signs. Exclusion criteria were as follows. 1) We excluded articles written in languages other than English. 2) We excluded articles outside the NH setting (i.e., acute settings, long-term care hospitals, community-dwelling settings). 

p.3 

Line 111-118

ï¾·  When "various" academic fields are mentioned, are they only from the health field or are other external fields included? It would be interesting to specify this.

Thank you for your comment. We specified included fields in this study. 

=> We included any type of research in designs in any academic field beyond health services (Nursing, Medicine, Dentistry, Psychology, Public health, Rehabilitation, Pharmacology, Nutrition). 

p.3 

Line 111-118

ï¾·  Please consider replacing "studies" with "designs".

Thank you for your comment. We replaced “studies” with “designs”. 

=>We also included studies with combined quantitative and qualitative designs.

p.3 

Line 115

ï¾·  A legend is needed in the table to clarify the circles, crosses and triangles. This version of the manuscript does not contain such data.

Thank you for your comment. We added a legend in table 1. 

=> Note. O: well-followed; â–²: followed but not address well; X: didn’t followed.

p.8 

Line 148

ï¾·  Please include the equations used for each database search as supplementary material to allow replication of the study.

Equations were listed in Supplement 1.

Attached 

ï¾·  Although it does not alter the final result of the selected articles, in the example equation, specifically in "("nursing home" [mh] OR "long term care" [all] OR "nursing home" [all])", may have led to articles unrelated to NHs (due to the inclusion of the term "long term care" together with the boolean operator OR).

Thank you for your comment. We stated process of selection articles in the result section. 

=>Figure 1 presents the degree of following PRISMA Checklist of the reviewed articles. In this review, we identified and screened a total of 1130 potentially relevant references identified through search of bibliographic databases. After deleting duplicates and not nursing home nor long-term care facility setting studies, the initial search yielded 73 publications. After screening abstracts, we excluded 32 references (30: systematic review only, 2: included only qualitative studies). Finally, 41 publications re-viewed and synthesized.

p.10 

Line 164-175

ï¾·  Please use PRISMA 2020 flow diagram for new systematic reviews which included searches of databases and registers only.

We changed from old PRISMA flow diagram to 2020 flow diagram (see figure1).

p.9

ï¾·  Replace "PRISMA guideline" with "PRISMA 2020 guideline".

Thank you for your comment. We replaced “PRISMA guideline" with "PRISMA 2020 guideline".

p.9 Line 157

ï¾·  It must be a typo, but both Figure 1 and Table 2 list 41 studies, while the text lists 39. Moreover, there is no mention of Figure 1 in the text.

Thank you for your comment. We mentioned of Figure 1 in the text and modified the number (39=>41). 

=> Figure 1 presents the degree of following PRISMA Checklist of the reviewed articles.

p.9

ï¾·  This part of the text should be restructured and amended. (1) Indicate the number of studies in number format or text, but do not mix both nomenclatures. (2) Line 140 and then lines 142-143 refer to studies using both PRISMA and MOOSE, so it seems to be repeated information (although, strangely, the number of studies differs in each sentence). (3) Similarly, the same information seems to be repeated for PRISMA in lines 139 and 142. (4) In lines 141-142, when RCTs are mentioned, it would be desirable to move this information to the end of the paragraph, as it is a different type of data from what is observed in the paragraph.

We restructured and edited as you recommended.

=> Table 2 presents the summary of the reviewed articles. 15 of 41 studies did not use any validated guidelines of MAs. The guidelines used in the remaining 26 studies are as follows; 17 studies used the only PRISMA guideline; 3 studies used the only MOOSE guideline; 3 studies used both the PRISMA and Cochrane handbook simultaneously; 3 studies used both the PRISMA and MOOSE guideline simultaneously.

p.10 Line 175-178

ï¾·  For each study listed in the table, please insert the corresponding number in the reference section.

c

Thank you for your comment. We added the corresponding number in the reference section in the table 2. 

p.10 

ï¾·  Please provide examples of measure-of-association in MAs.

Thank you for your comment. We added the information and examples of measure-of-association in MAs. 

Measure-of-association represents various coefficients (including bivariate correlation coefficients and regression coefficients) that measure the strength and direction of relationships between variables [73]. These intensity or association measures can be explained in several ways depending on the analysis. Measure-of-association in MAs includes analyzing using studies analyzed using methods such as correlation, regression, and structural-equation modeling between variables [12]. 

p.17 Line 193-198

ï¾·  Consider moving this information to Discussion.

Thank you for your comment. We moved the following contents to the discussion section. 

p.18 241~268

ï¾·  This information is either confusing or I have misunderstood it. The PRISMA 2020 statement explains that it can be applied to different types of systematic reviews (including those that only review other systematic reviews).

Thank you for your comment. We modified the sentences. 

=> However, some reviewed studies used the PRISMA (which should not be designed for observational studies) despite not having RCTs in their studies, [34–36,21]. Unintentional inadvertence may occur when reporting reviews without adhering to valid review guide-lines. For example, the APA MARS is a good guideline for association and relational MA in addition to non-RCT systematic reviews and MAs in NH fields. The APA Working Group on Journal Article Reporting Standards developed the APA MARS in 2008 after the group synthesized the Quality of Reporting of MA statement, PRISMA, MOOSE, and Potsdam Consultation on MA (APA, 2008). The Newcastle-Ottawa scale has also been used when reviewing observational studies including case-control, prospective, and cohort studies [75]. This scale has three areas (participants of study, comparison between case/controls, and evaluation of outcomes), which are linked with quality concept and recommended to have at least two reviewers as the review offers quite subjective characteristics [75]. The MOOSE is a reporting guide for MAs of observational studies in epidemiology. The MOOSE improves reporting MAs of observational studies [76]. The Strengthening the Re-porting of Observational Studies in Epidemiology (STROBE) Initiative developed recommendations on what should be included in an accurate and complete report of an observational study. STROBE was developed to improve the reporting quality of observational studies [77]. Regarding quality-assessment tools, one of the examined studies used Consolidated Standards of Reporting Trials (CONSORT) [4] to measure methodological quality for individual studies included in MA research. However, CONSORT is a set of recommendations for how to report on individual studies of randomized trials, not actually a quality-assessment tool for MAs to establish internal validity. Even though meta research is considered the highest scientific level of evidence, many potential sources of biases exist [78] including selection, performance, attrition, and detection bias, which affect internal validity. Therefore, scholars must use valid and standardized, methodological quality-assessment tools for individual studies included in MA research to ensure the quality of synthesized results.  

p.18 241~268

ï¾·  Some studies have assessed quality using ROB instead of ROB 2 because were published before ROB 2 was launched. Please specify which studies were evaluated using ROB when ROB 2 was already available.

Thank you for your comment. We specified which studies were evaluated using ROB or ROB 2. 

p.18 269-271

ï¾·  "d" for Odds Ratio or Cohen's d?

Thank you for your comment. “d” is Odds Ratio in this sentence. We added it. 

=>Effect size usually consists of d (odds ratio) for experimental designs and R for correlational studies [8].

p. 19 Line 275

ï¾·  Many of the future recommendations focus on researchers. However, scientific journals can also contribute to the publication of research with higher methodological quality (requirement to use reporting guidelines from EQUATOR Network, prior publication of the protocol in prospective registers such as PROSPERO...). Please add these contributions.

The intervention complexity of studies (intervention number and the interaction among interventions in a system’s structure) should be considered in the future as a mixture of interventions should be applied to NH residents with a more accurate intervention estimation [41]. The interventions which were conducted to improve residents’ physical, psychological, and cognitive functions include fall prevention, nutrition, oral care, ambulant, feeding, exercise, and so on. 

p. 19 Line 285-299

Reviewer 2 Report

Abstract

  • Please mention databases searched, as well as search period.
  • Results and conclusion are poorly stated. Please quantify some results. Then, please discuss your important findings, and then allude to limitations, and finally state strong and useful recommendations.

Methods

  • What were your exclusion criteria?
  • You included only English papers and this is a big limitation.
  • Please mention Kappa statistics value regarding agreement between the reviewers.

Results

  • You included 39 papers, but it is 41 in PRISMA! How is this inconsistency justified!?

Discussion

  • Please state your strengths and limitations separately.
  • Lacking PROSPERO registration number and excluding non-English papers were some of your limitations.

Author Response

December, 15, 2021

Executive Editor, International Journal of Environmental Research and Public Health

RE: Revision of Manuscript ID: ijerph-1492274 entitled "A Methodological Quality Evaluation of Meta-Analyses on Nursing Home”

Research: Overview and Suggestions for Future Directions"

To whom it may concern:

Attached please find an electronic copy of a first revision of Manuscript ID: ijerph-1492274 entitled "A Methodological Quality Evaluation of Meta-Analyses on Nursing Home,” reviewed for publication in the International Journal of Environmental Research and Public Health. In your editorial decision letter, you observed that the reviewers found considerable merit in the manuscript, but a number of issues required attention. You invited a revision and resubmission of the manuscript. We found your comments and those provided by the reviewers to be extremely helpful, and we hope you will find the revised manuscript to be substantially improved. Below, we provide a detailed point-by-point description of our response to each issue the reviewers raised. 

We believe we have addressed each issue and hope you find the revisions to be satisfactory. We look forward to hearing from you. Thank you for your time and consideration.

Abstract

ï¾· Results and conclusion are poorly stated. Please quantify some results. Then, please discuss your important findings, and then allude to limitations, and finally state strong and useful recommendations.

Systematically, we revised the results and discussion section. 

(4) Results: The studies primarily fell into the following categories: medicine (17/41), nursing (7/41), and psychiatry or psychology (6/41). 36.6% of reviewed studies did not use any validated MA guidelines. The lowest correctly reported PRISMA guideline item was protocol and registration (14.6%) and more than 50% of articles did not report risk of bias. Moreover, 78.0% of studies did not describe missing reports of effect size formula; (5) Discussion: NH researchers must follow appropriate and updated guidelines for their MAs to provide validated reviews and consider statistical issues such as complexity of interventions, proper grouping, and scientific effect-size calculations to improve the quality of study. Future quality review studies should investigate more diverse studies.

p.1 Line 16-28

Methods

ï¾· You included only English papers and this is a big limitation.

Thank you for your comment. We included the point (including only articles published in English) in the limitation section. 

=>included only articles published only in English. Future studies should include more diverse research around the world to represent citizens in the world.

p.20 Line 310

Methods

ï¾·  

Thank you for your comment. We mentioned Kappa statistics value regarding agreement between the reviewers. 

=>The kappa statistics value showed a high degree of agreement between the three reviewers in terms of their decisions about the relevance of studies (kappa value =0.89).

p.4 Line 140-141

Results

ï¾· You included 39 papers, but it is 41 in PRISMA! How is this inconsistency justified!?

Thank you for your comment. We are sorry to confusing you. We added 41 papers and revised all texts.

All page 

Reviewer 3 Report

I think, this is a Critical Appraisal of MAs on Nursing Home, of published articles. 

It´s important to obey the guidelines in research and in the different kind of studies. Here, we are talking about MAs, it is important to use PRISMA and to do the correct steps, including to detect biases in order don´t select that article with biases, because at the end we will have incorrect results.

You found low level in statistical issues.

I would like to listen a comment about the I 2, (heterogeneity test), that give us an idea of big heterogeneity when this % is high.

Author Response

December, 15, 2021

Executive Editor, International Journal of Environmental Research and Public Health

RE: Revision of Manuscript ID: ijerph-1492274 entitled "A Methodological Quality Evaluation of Meta-Analyses on Nursing Home”

Research: Overview and Suggestions for Future Directions"

To whom it may concern:

Attached please find an electronic copy of a first revision of Manuscript ID: ijerph-1492274 entitled "A Methodological Quality Evaluation of Meta-Analyses on Nursing Home,” reviewed for publication in the International Journal of Environmental Research and Public Health. In your editorial decision letter, you observed that the reviewers found considerable merit in the manuscript, but a number of issues required attention. You invited a revision and resubmission of the manuscript. We found your comments and those provided by the reviewers to be extremely helpful, and we hope you will find the revised manuscript to be substantially improved. Below, we provide a detailed point-by-point description of our response to each issue the reviewers raised. 

We believe we have addressed each issue and hope you find the revisions to be satisfactory. We look forward to hearing from you. Thank you for your time and consideration.

Reviewers' comments (Reviewer 3)

Response to the comments

Page and Lines

ï¾· Lacking PROSPERO registration number and excluding non-English papers were some of your limitations.

ï¾· I would like to listen a comment about the I 2, (heterogeneity test), that give us an idea of big heterogeneity when this % is high.

Thank you for your comment. We stated this point in the text. 

=>Scholars used H (1/41), Q (18/41) and I2 inconsistency (29/41) statistics for heterogeneity tests. Heterogeneity means the degree of differences in the results of each single research finding (Walker et al., 2008). Through MAs, scholars calculate the heterogeneity index so they can understand the primary factors that impact individual studies’ effect sizes. 

p.18 Line 213-217

Round 2

Reviewer 1 Report

Thank you for improving the manuscript. Just a few more suggestions based on the added text:

  • Lines 148-149. Figure 1 could be placed after the paragraph in which it is mentioned (lines 166-167).
  • Line 209. Please change "scholars" to "researchers", as the word is repeated a few lines earlier.
  • Lines 294-298. All comments on the EQUATOR Network need to be deleted, as they are indirectly dealt with in lines 232-260 (new content added) and have a different purpose than PROSPERO. The paragraph should read: "...quality using PROSPERO. The protocol registration...".

Best regards.

Author Response

I totally agree with your  comments, and got the professional English editing from the  one of the editing services listed at 
https://www.mdpi.com/authors/english. Thank you again for your precious reveiw.

Reviewer 2 Report

Thank you for the revisions.

Author Response

December, 18, 2021

Executive Editor, International Journal of Environmental Research and Public Health

RE: Revision of Manuscript ID: ijerph-1492274 entitled "A Methodological Quality Evaluation of Meta-Analyses on Nursing Home”

Research: Overview and Suggestions for Future Directions"

To whom it may concern:

Attached please find an electronic copy of a first revision of Manuscript ID: ijerph-1492274  entitled  "A Methodological Quality Evaluation of Meta-Analyses on Nursing Home,” reviewed for publication in the International Journal of Environmental Research and Public Health. In your editorial decision letter, you observed that the reviewers found considerable merit in the manuscript, but a number of issues required attention. You invited a revision and resubmission of the manuscript. We found your comments and those provided by the reviewers to be extremely helpful, and we hope you will find the revised manuscript to be substantially improved. Below, we provide a detailed point-by-point description of our response to each issue the reviewers raised.

We believe we have addressed each issue and hope you find the revisions to be satisfactory. We look forward to hearing from you. Thank you for your time and consideration.

  • Lines 148-149. Figure 1 could be placed after the paragraph in which it is mentioned (lines 166-167).
  • Yes, we placed the Figure 1 after the paragraph.
  • Line 209. Please change "scholars" to "researchers", as the word is repeated a few lines earlier.
  • Yes, we changed it.
  • Lines 294-298. All comments on the EQUATOR Network need to be deleted, as they are indirectly dealt with in lines 232-260 (new content added) and have a different purpose than PROSPERO. The paragraph should read: "...quality using PROSPERO. The protocol registration...".
  • I totally agree with you. I removed it for the better clarity. .